# Structural characterization of ice XIX as the second polymorph related to ice VI

Tobias M. Gasser [1], Alexander V. Thoeny [1], A. Dominic Fortes [2] & Thomas Loerting [1 ✉]

Ice polymorphs usually appear as hydrogen disorder-order pairs. Ice VI has a wide range of thermodynamic stability and exists in the interior of Earth and icy moons. Our previous work suggested ice β-XV as a second polymorph deriving from disordered ice VI, in addition to ice XV. Here we report thermal and structural characterization of the previously inaccessible deuterated polymorph using ex situ calorimetry and high-resolution neutron powder diffraction. Ice β-XV, now called ice XIX, is shown to be partially antiferroelectrically ordered and crystallising in a $\sqrt{2}\times\sqrt{2}\times1$ supercell. Our powder data recorded at subambient pressure fit best to the structural model in space group $P\bar{4}$. Key to the synthesis of deuterated ice XIX is the use of a DCl-doped $D_2O/H_2O$ mixture, where the small $H_2O$ fraction enhances ice XIX nucleation kinetics. In addition, we observe the transition from ice XIX to its sibling ice XV upon heating, which proceeds via a transition state (ice VI$^{\ddagger}$) containing a disordered H-sublattice. To the best of our knowledge this represents the first order-order transition known in ice physics.

[1] Institute of Physical Chemistry, University of Innsbruck, Innsbruck, Austria. [2] ISIS Neutron and Muon Facility, Rutherford Appleton Laboratory, Harwell Science and Innovation Campus, Chilton, Oxfordshire OX11 0QX, UK. ✉email: thomas.loerting@uibk.ac.at

Hexagonal ice (ice $I_h$), familiar to most of us from snow or the freezer, is a geometrically frustrated ice. While the oxygen atoms fully occupy crystallographic sites in the molecular framework, the hydrogen atoms are disordered over partially-occupied sites. Yet, the Bernal-Fowler rules[1], also known as the ice rules, are obeyed, in the sense that (i) the crystal is composed of $H_2O$ molecules and (ii) there is always a single hydrogen atom between two oxygen atoms, so that a space-filling hydrogen-bonded network is formed. Following these rules many different configurations are allowed, where the total number of possible microstates gives rise to a configurational entropy $S_{conf}$ of approximately $S_{conf} = R*ln(3/2)$ known as Pauling entropy[2]. Below the order-disorder temperature $T_{o-d}$ thermodynamics requires a fully ordered phase to be more stable than this frustrated ice. In this phase both H and O atoms are ordered, so that the configurational entropy is zero. How the H atoms order and the net dipole moment of the ordered structure are a challenge to predict accurately. Quite often the ordered ice is not accessible through simple cooling. In case of ice $I_h$ $T_{o-d}$ was found to be very low, at 72 K[3–5] – and yet, even upon cooling ice $I_h$ slowly below 72 K it does not order. The times required for order to develop are simply beyond laboratory time scales. Only in 1972 Kawada[3] achieved the transition, where applying two key concepts allowed access to the ordered ice[6]: (i) nucleation of the ordered phase by keeping the sample for a few days about 10 K below $T_{o-d}$, (ii) accelerated growth of the ordered ice phase through the use of extrinsic defects just below $T_{o-d}$. Specifically, Kawada was the first to use KOH doping, which generates an ionic OH- defect (violating ice rule (i)) together with a Bjerrum-L-defect (violating ice rule (ii))[7]. Such defects are now known to accelerate the reorientational dynamics by up to five orders of magnitude[8]. The ordered phase related to ice $I_h$ that needs growth times of a few days is called ice XI. Ice XI represents the ground state of ice at 0 K and ambient pressure, not ice $I_h$ or ice II as previously believed[9]. Its ferroelectric ordering determined from single crystal neutron diffraction data[6] and molecular simulations[10] has been challenged and subsequently claimed to be antiferroelectric[11]. The same issues arise for a plethora of high-pressure ice phases. For instance, disordered ice V and ice XII, could only be transformed to their ordered counterparts, ice XIII and ice XIV, through the use of HCl doping[12]. HCl doping introduces an ionic $H_3O^+$ defect, together with a Bjerrum-L-defect. In these cases $T_{o-d}$ is 112 ± 3 K[8,12–15] and 102 ± 3 K[12,14,16–18], respectively. Their growth takes place on the time scale of hours rather than days because of the higher $T_{o-d}$. Undoped or KOH-doped high-pressure ices do not order sufficiently fast, for reasons that are yet not fully understood. The ice III/IX and ice VII/VIII order-disorder pairs have $T_{o-d}$s of 170 K and 273 K[7], respectively – high enough so that the use of dopants is no longer necessary to achieve ordering.

A particularly challenging and interesting case is disordered ice VI. Ice VI exists as a thermodynamically stable phase in a broad pressure range between 0.6 and 2.2 GPa and at temperatures up to 355 K[7]. It is found naturally in Earth's mantle[19] and in the interior of icy moons such as the Galilean satellites[20]. Partially antiferroelectrically ordered ice XV forms from ice VI upon cooling, again, only in the presence of acid doping[21]. In earlier work, without the use of a dopant, only a very slow ordering process could be inferred[22]. Its $T_{o-d}$ has been determined to be 129 ± 2 K[21,23–26]. Yet, much more complex kinetics than in other ice phases has been recognized, with a low-temperature tail extending to as low as 100 K[27]. This hints towards the existence of a competing ordering process, even though not recognized as such at the time[27]. The transition is also clearly affected by the choice of pressure. The ordering process to yield ice XV seems to be impeded at high-pressures, but alleviated at ambient pressure.

In 2018 some of us have claimed that the ordering process to ice XV is not only slowed down severely near 2 GPa, but instead a transition to a differently-ordered phase takes over[25]. This ordering process leads to a phase called ice β-XV by us, where $T_{o-d}$ for ice VI/β-XV was shown to be about 26 K below the $T_{o-d}$ for ice VI/XV[25]. In other words, the low temperature tail recognized first by Shephard & Salzmann[27] originates from the transition of remaining ice VI to ice β-XV. Supporting this hypothesis, calorimetry experiments have shown that two disordering processes take place upon heating ice β-XV[25]. Both ice β-XV and ice XV only release a small fraction of the entropy expected for the transition from a fully ordered to a fully disordered state. In other words, calorimetry suggests both polymorphs to be weakly ordered variants of ice VI[25]. For ice XV this claim has already been crystallographically demonstrated[21], while for ice β-XV crystallographic analysis has so far remained elusive. Another key experiment to make the case for the presence of two distinct ordered phases is dielectric relaxation spectroscopy, which shows the activation energy for dielectric relaxation as 45 kJ mol$^{-1}$ in ice β-XV, but only 27 kJ mol$^{-1}$ in ice XV[25]. Raman spectroscopy reveals clear differences between ices VI, XV and β-XV, e.g., in the librational and decoupled OH-stretching regions[26]. Some of us have previously argued that an unfavourably large kinetic isotope effect hampers formation of deuterated ice β-XV[25], the deuterated analogue being required for accurate determination of hydrogen site ordering using neutron powder diffraction. The same slow cooling process that allows $H_2O$ ice β-XV to form from $H_2O$ ice VI just does not allow $D_2O$ ice β-XV to form from $D_2O$ ice VI. The lack of an experimental crystal structure for ice β-XV has led to widespread speculations about its nature. Whereas Rosu-Finsen & Salzmann have argued for ice β-XV to be in fact a disordered state, in which the H atoms are immobile, i.e., a deep glassy state[28,29], some of us have speculated about a crystalline phase with a ferroelectric nature[25].

We here show that in fact growth kinetics of deuterated ice β-XV is not at the origin of the large isotope effect. Instead, nucleation kinetics of deuterated ice β-XV in ice VI is the limiting step. Our key result is that nucleation kinetics of deuterated ice β-XV is significantly enhanced by adding small amounts of $H_2O$. Adding only 0.5% of $H_2O$ to pure $D_2O$ allows for the formation of $D_2O$ ice β-XV. Using samples containing a small fraction of water, and DCl as dopant, we provide neutron powder diffraction data and determine its crystal structure. For this reason ice β-XV can now be renamed as ice XIX, following the recent discovery of superionic ice XVIII[30]. The ice XIX to ice XV transition incurred upon heating represents, to the best of our knowledge, the only order-to-order transition in the H-sublattice known in any kind of water ice.

## Results

**Calorimetry: nucleation and growth of $D_2O$ ice XIX.** Both ice XV and ice XIX samples in this work were made through isobaric cooling of ice VI samples, where DCl doping is used to enhance reorientation dynamics. Ice XIX is obtained at 1.8 GPa upon very slow cooling of ice VI from 255 K to 77 K. Ice XV, used as a reference for the neutron diffraction experiments, is obtained upon cooling doped ice VI slowly at 50 mbar from 135 K to 70 K (see Methods section for more details). Figure 1 shows ambient pressure calorimetry scans for $H_2O$, $D_2O$ and mixed $D_2O/H_2O$ samples. For pure $H_2O$ samples the cooling procedure at 1.8 GPa is identical to the one shown to result in the formation of ice β-XV in our previous work[25]. The calorimetry trace for pure $H_2O$ (top trace in Fig. 1a) shows two well separated endotherms, where the first one indicates disordering of ice β-XV and the second one indicates disordering of ice XV, resulting in ice VI.

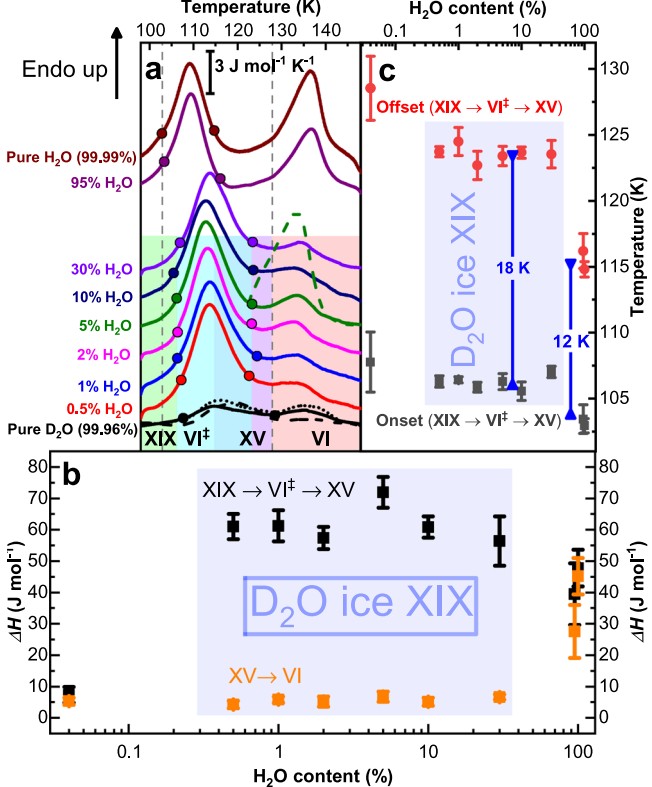

**Fig. 1 Calorimetric characterization of hydrogen order. a** Differential scanning calorimetry traces of ice XIX samples with different $D_2O/H_2O$ ratios recorded at a heating rate of 10 K min$^{-1}$. The two endotherms indicate first the ice XIX → VI$^{\ddagger}$ → XV and second the ice XV → VI transition. All full lines were recorded on samples slow-cooled at 1.8 GPa (see Methods). Dashed lines mark heating scans of ice XIX annealed at 1.8 GPa and 106 K (black dashed line, pure $D_2O$) and annealed at ambient pressure and 120 K (green dashed line, 5% $H_2O$). Heating scan of ice XIX after very slow cooling at 1.8 GPa is shown as dotted black line (pure $D_2O$). Onset and offset points for the first transition are marked by full circles. **b** Enthalpy changes associated with the two transitions (black squares: XIX → VI$^{\ddagger}$ → XV; orange squares: XV → VI). **c** Onset (black squares) and offset points (red circles). Error bars in **b** and **c** reflect both reproducibility and ambiguities in determining the points based on the tangent method. The width of the transition at 10 K min$^{-1}$ is indicated through the blue double arrow.

As indicated in Fig. 1b both endotherms are approximately the same size of 50 J mol$^{-1}$, in agreement with our earlier work (Fig. 2 in ref. [25]). Considering the difference of 25 K in $T_{o-d}$ for these two endotherms this corresponds to a loss of $S_{conf}$ of 0.46 and 0.35 J mol$^{-1}$K$^{-1}$ in pure $H_2O$ samples, respectively. For pure $D_2O$ the exact same procedure results in the trace depicted at the bottom of Fig. 1a. In spite of the large magnification used in Fig. 1a (see scale bar) there are hardly any features different from the baseline. Yet, two very subtle endotherms (close to the resolution limit of the instrument) can be identified, both of which are less than 10 J mol$^{-1}$. That is, the bulk of the sample had remained disordered ice VI upon cooling under pressure. This had prevented neutron diffraction work on the crystal structure before the present study. In order to resolve the issue we have first attempted to reduce the cooling rate significantly below 3 K min$^{-1}$, which is the rate that successfully produces $H_2O$ ice β-XV. The lowest cooling rate employed for $D_2O$ samples was 0.1 K min$^{-1}$, thereby providing 13 h for the sample to order upon cooling. Even reduction of the cooling rate by this factor of 30 has

not changed the picture, where still barely any ordering takes place upon cooling $D_2O$ ice VI (see black dotted trace in Fig. 1a). Furthermore, we have attempted to isothermically keep the sample below the $T_{o-d}$ of ice β-XV for 25 h. As mentioned above, this strategy has proven to be successful to generate ice XI from ice $I_h$. The black, dashed calorimetry trace at the bottom of Fig. 1a, however, shows that also this strategy has not resulted in $D_2O$ ice β-XV.

The mere fact that reducing the cooling rate still does not increase the order within the ice VI sample points against phase growth as the issue. Instead nucleation of $D_2O$ ice β-XV is inhibited and the limiting factor that needs to be tackled to prepare the deuterated polymorph. We know from our previous work that $H_2O$ ice β-XV successfully nucleates within ice VI[25]. Thus, we have attempted to enhance $D_2O$ ice β-XV nucleation kinetics by adding $H_2O$. The result of deliberately adding $H_2O$ to $D_2O$ samples is summarized in Fig. 1a, for samples containing between 0.5% and 95% of $H_2O$. Already 0.5% makes a tremendous difference. The disordering endotherm near 100 K suddenly appears with an enthalpy of 61 J mol$^{-1}$. Repeating the experiment with 1%, 2%, 5%, 10% or even 30% added $H_2O$ barely changes this picture. Only for the 95% $H_2O$ sample the calorimetry traces of pure $H_2O$ are restored, which suggests that our strategy has been successful. This is confirmed when inspecting isotope effects on the latent heat and disordering temperature. The enthalpy associated with disordering of $D_2O$ ice β-XV is about 64 ± 8 J mol$^{-1}$ as compared to 45 ± 5 J mol$^{-1}$ for the case of $H_2O$ ice β-XV – a typical isotope effect for endotherms. The onset point for the $D_2O$ sample shifts to higher temperatures by about 3 K – again a quite typical isotope effect[18,31]. That is, the DSC traces suggest that $D_2O$ ice β-XV indeed does form in all mixtures containing from 0.5% to 30% $H_2O$. For this reason these compositions are coloured in blue and marked $D_2O$ ice XIX in Fig. 1b/c. From the enthalpy of disordering and the $T_{o-d}$ of 107 K an entropy change of 0.60 ± 0.07 J K$^{-1}$ mol$^{-1}$ results. This corresponds to 18 ± 2% of the Pauling entropy.

In the Supplementary Discussion we show why the second endotherm (ice XV → ice VI) in Fig. 1a is much smaller in deuterated than protiated samples. In brief, this reflects that the formation of ice XV is much slower in deuterated samples, as seen from the larger width of the first endotherm (18 K vs. 12 K, see blue arrow in Fig. 1c). This implies that the transition from ice XIX to ice XV proceeds through a disordered transition state, called ice VI$^{\ddagger}$ here.

**Neutron powder diffraction: comparing ice VI, XV and ice XIX.** Let us now turn to the neutron powder diffraction measurements of the deuterated samples on the High Resolution Powder Diffractometer, HRPD, at the ISIS spallation neutron source. We recorded data out to 10 Å and used the highest-resolution data in the backscattering banks at $d$-spacings from 0.65 to 2.60 Å for structure refinements. Figure 2 compares the powder diffraction patterns for undoped ice VI, ice XV and ice β-XV, all recorded on HRPD. The ex situ neutron diffraction pattern of ice VI recorded at 70 K under ~50 mbar of He exchange gas matches very well with the known pattern, as measured in situ at 225 K and 1.1 GPa by Kuhs et al.[32] or ex situ by Salzmann et al.[21] on the GEM instrument. It also matches the tetragonal $P4_2/nmc$ structure deduced by Kamb in 1965 from X-ray measurements[33]. Our refined ice VI structure contains comparatively short O-D distances, ranging from 0.917 to 0.936 Å, which is a consequence of orientational disorder leading to static distortions as outlined by Kuo and Kuhs[34]. The same effect also appears for other disordered ices, such as ices III and

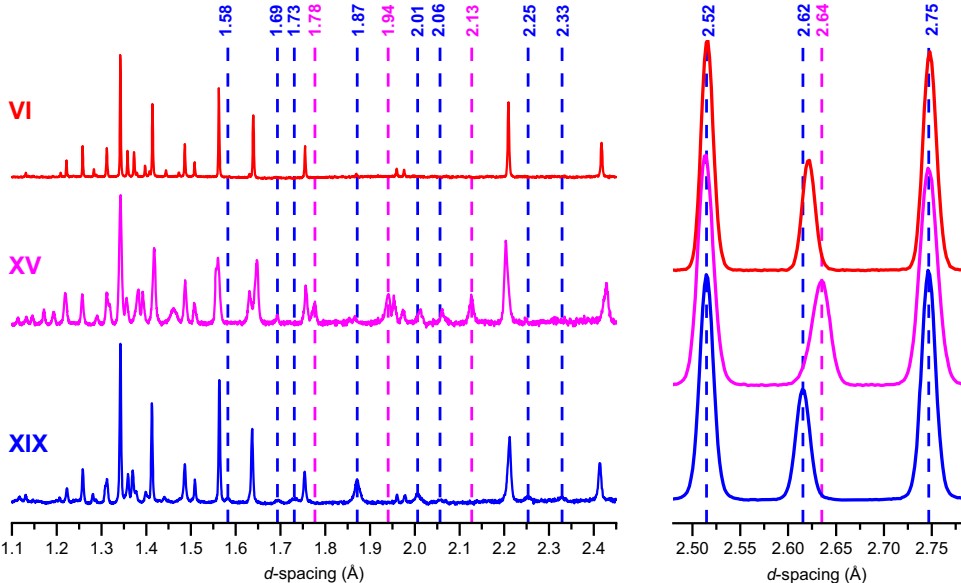

**Fig. 2 Comparison of ice XIX, ice XV and ice VI neutron powder diffraction patterns.** Baselines were corrected with an 8 pt spline. Selected features of the ice XIX and ice XV diffractograms are highlighted by vertical, dashed blue and magenta lines, respectively. Note the sharp peaks in ice VI and the broader peaks in ices XV and XIX—this hints at particle size broadening originating from very small partially ordered domains. The diffractograms on the left were acquired in the highest resolution backscattering banks (2θ = 165 ± 11°) whilst the three peaks shown on the right were recorded in the medium-resolution detectors at 2θ = 90 ± 10°.

V[35] or VII[32]. The ice XV pattern recorded from a sample recooled from 135 K with a cooling rate of 0.4 K min$^{-1}$ at 50 mbar matches nicely with the pattern recorded by Salzmann et al.[21]. Compared to ice VI, new Bragg peaks at 1.775, 1.938, 2.013, 2.066 and 2.129 Å are found in the ice XV pattern[21]. The ice β-XV pattern recorded from a sample containing 5% $H_2O$ is depicted at the bottom of Fig. 2. This pattern is different from both the ice XV and the ice VI patterns. Eight Bragg reflexes at d-spacings between 1.5 and 2.4 Å are marked by blue dashed lines that appear in ice β-XV, but not in ice VI. The 1.583, 1.734, 2.254 and 2.326 Å features also do not appear in ice XV. The 1.775, 1.938 and 2.129 Å features characteristic of ice XV on the other hand do not appear in ice β-XV (see dashed, magenta lines in Fig. 2). Also the shift of the intense feature from 2.64 Å in ice XV to 2.62 Å in ice β-XV is noteworthy. That is, the ice β-XV pattern is different from ice XV and ice VI, in spite of sharing the same kind of oxygen network. Note that the Bragg peaks observed in backscattering from ice VI are substantially sharper than in either ice XV or ice β-XV. This appears to be the result of particle-size broadening in the latter two phases, most likely due to the development of many small domains ordering upon cooling ice VI. Below we show the behaviour upon heating in HRPD and the refinement of the recorded data, which makes the case for ice β-XV being renamed to ice XIX.

**Order-order transition ice XIX → ice XV based on neutron powder diffraction.** The phase behaviour upon heating ice XIX from 70 to 132.5 K is demonstrated in Fig. 3 for seven selected marker regions. To aid the eye blue, pink and red dashed lines are used to indicate marker bands for ice XIX, ice XV and ice VI, respectively. Let's first inspect the 1.69 and 1.73 Å features characteristic of ice XIX (blue in Fig. 3). They are well resolved at 70 K, and still well resolved at 100 K. At 102.5 K these two features shrink in intensity and then disappear at 105 K. At 105 K the pattern is practically featureless, as is the case in ice VI. Upon continued heating the feature at 1.69 Å reappears (slightly shifted), but the one at 1.73 Å does not. This trend continues until

120 K, where the 1.69 Å reflex is nicely developed. This pattern is characteristic of ice XV (pink in Fig. 3). Upon further heating this peak disappears again, and has fully vanished at 132.5 K, at which temperature ice VI (red in Fig. 3) is developed. At all other temperatures the patterns are depicted in grey to indicate that at least two phases contribute to the diffraction signal. The shift of the reflex at 1.22 Å is also quite revealing. Compared to 70 K the peak first upshifts at 102.5 K, close to its position in ice VI. That is, ice XIX first disorders, producing ice VI‡. On continued heating the feature downshifts again, finally reaching the ice XV position at 120 K. That is, between 102.5 K and 120 K ordering of ice XV takes place starting from transient ice VI‡. In the diffraction experiment the ordering reaches completion because many hours are available for the process, as opposed to the calorimetry experiment in Fig. 1a providing only 2.5 min. Above 120 K the position upshifts again, reaching the ice VI position a second time at 132.5 K. Such thermal behaviour could not be explained without phase changes, and without invoking an ice VI‡ transition state in the order-order transition from ice XIX to ice XV. The disappearance of ice XIX can be best seen at 1.87 Å. This comparably intense feature disappears as ice XV and ice VI form. The feature can be seen at least up to 102.5 K, maybe even 105 K, above which ice XIX has fully disappeared in the diffraction experiment. This is fully consistent with the onset temperature of 107 ± 1 K obtained in the calorimetry experiment in Fig. 1c, where the onset in the fast heating scans marks the full conversion in the slow heating experiment. The disappearance of the ice XIX features at 2.01, 2.05, 2.25 and 2.33 Å is fully consistent with this. Figure 4 shows changes in the c-axis upon transforming ice XIX to ice XV. It is immediately evident that ice XIX features a shorter length than ice VI, whereas ice XV shows a longer one. Furthermore, ice XIX (449.2 Å$^3$, see Table 1) also features a smaller unit cell volume than ice XV (450.6 Å$^3$)[21]. This explains why high pressure favors the formation of ice XIX over ice XV and vice versa at low pressure. At about 107 K the c-axis length obtained in the heating scan of ice XIX crosses the c-axis length of undoped ice VI. This is exactly where the disordered transition state ice VI‡ is indicated in the calorimetry and neutron

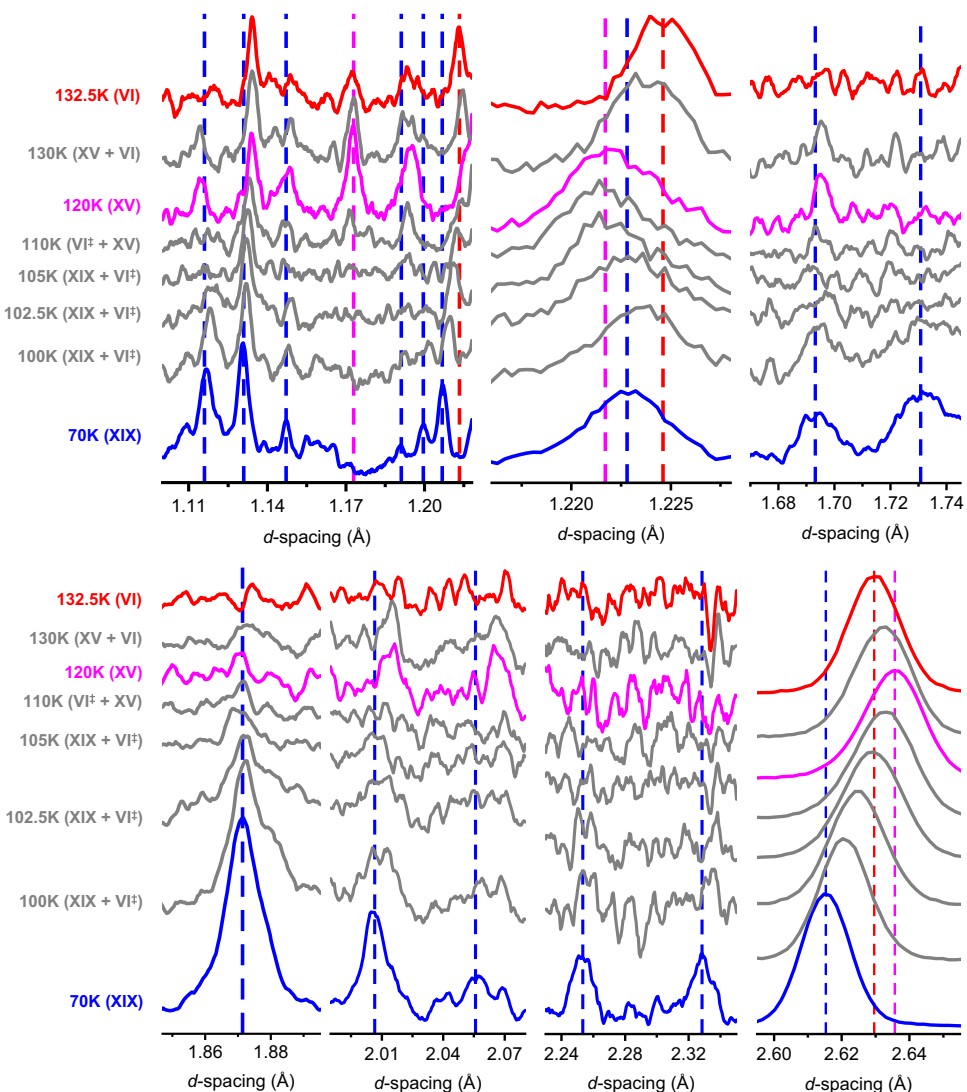

**Fig. 3 Neutron diffraction of the order-order-disorder transition.** Heating of a $D_2O$ ice XIX sample in the High Resolution Powder Diffractometer (HRPD), revealing transitions to ice XV (120 K) (via ice VI‡) and to ice VI (132.5 K). Data in panels 1, 3, 4, 5 and 6 were smoothed using a 13 pt Savitzky-Golay. Blue, purple and red dashed lines mark important Bragg peaks for ice XIX, XV and VI, respectively.

diffraction experiments described above. Above 130 K the *c*-axis length in ice XV then drops back from 5.805 to 5.792 Å, i.e., the value of ice VI. Thus, the transition sequence ice XIX → ice VI‡ → ice XV → ice VI is clearly evidenced from these heating experiments and marked accordingly in Fig. 1a using the coloured boxes.

**Refinement of ice XIX from neutron powder data**. The issue of refining the ice XIX structure was simultaneously also tackled by Yamane et al.[36]. These authors have studied the ice XIX structure in situ at 1.6 GPa using small volume samples, whereas we have studied the ice XIX structure ex situ at about 50 mbar using gram quantities of ice XIX. Consequently, their peak intensities are smaller than ours and suffer from significant peak overlapping with reflections from the high-pressure cell. Furthermore, they have not added a small amount of $H_2O$, which leads us to think that the fraction of ice XIX in their sample is smaller than in ours due to adverse nucleation kinetics for $D_2O$ ice XIX.

The data for pure ice VI were refined by the Rietveld method in GSAS/Expgui[37], starting from the structural model of Kuhs et al.[32]. This yielded a fit with $\chi^2 = 2.307$ and weighted Rietveld

powder statistic $w$Rp = 1.50 %. Lattice parameters for ice VI at 70 K were found to be $a = 6.242485(11)$ Å, $c = 5.770048(22)$ Å, $V = 224.851(1)$ Å$^3$. This structure refinement formed the basis for deriving models to test against the ice XIX data. One observes in the doped ice sample that many peaks permitted by the $P4_2/nmc$ space group of ice VI, but which have negligible or zero intensity in ice VI, adopt a measurable intensity in ice XIX, albeit being very broad. Examples include the 212 and 311 peaks at $d = 2.00$ and 1.868 Å respectively (marked * in Fig. 5). In addition to this, there are a few exceptionally weak and broad peaks that are not permitted by the cell metric of ice VI, which can only be accounted for by the development of a supercell. Most notable are the super-lattice peaks at 2.326 and 2.254 Å, respectively (marked † in Fig. 5). Indexing of the ice XIX powder diffraction data, including the weak super-lattice Bragg peaks was done using DICVOL06[38], from which a √2×√2×1 super-cell with twice the unit cell volume of ice VI is obtained. Despite the high resolution of HRPD the broad peaks characteristic of ice XIX preclude unambiguous detection of any evidence for orthorhombic or monoclinic distortions of the super-cell by the splitting of Bragg peaks or even the development of noticeable shoulders. Whole-pattern profile fitting using the LeBail method inevitably

results in better fits to the data with lower symmetry cell metrics, simply by virtue of the extra degrees of freedom, when the same result might equally be achieved with a higher symmetry cell and a more complex description of the peak shapes. On this basis, possible distortions of the unit cell to lower symmetry monoclinic or triclinic cells were precluded from further consideration.

In total almost 2000 possible structure models are feasible for the supercell of double the ice VI unit cell[39]. This overwhelming number of possible structures could be narrowed down by investigating each of the possible subgroups of the √2×√2×1 super-cell regarding its compatibility with the ice VI-oxygen

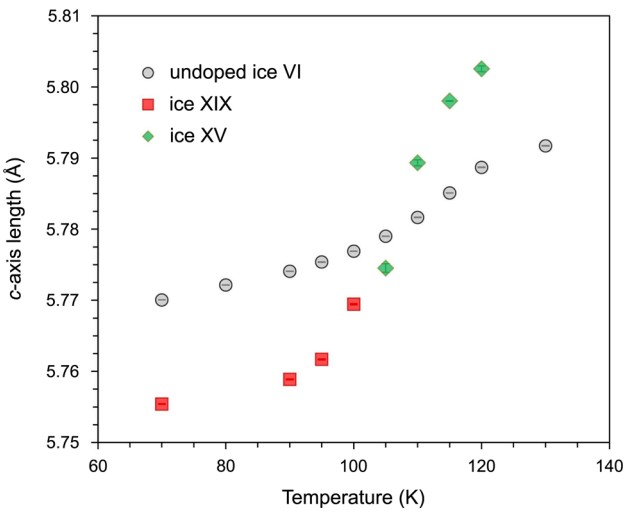

**Fig. 4 Unit cell data.** Change in the $c$-axis length upon heating ice XIX (red squares and green diamonds, obtained from Rietveld refinement of the data set shown in Fig. 3) compared with a heating scan of undoped ice VI (circles).

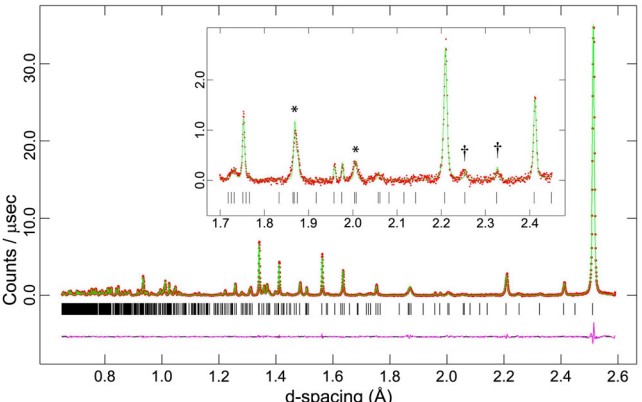

**Fig. 5 Rietveld refinement of $P\bar{4}$ model.** Neutron powder diffraction data for ice XIX (red circular symbols) and the fit calculated from the refined $P\bar{4}$ model (green line), with the background subtracted. The difference between the model and data is represented by the purple line underneath the diffraction pattern, and the positions of the Bragg peaks are indicated by vertical black tick marks. The inset shows the region between 1.7 and 2.45 Å $d$-spacing where peaks appear that are permitted by the ice VI cell metric but have little or no intensity in ice VI (*) and where super-lattice peaks are observed (†).

**Table 1 Refined structure using the $P\bar{4}$ model.**

| Atom | Wyckoff position | x | y | z | Occupancies | $U_{iso}$ |
|---|---|---|---|---|---|---|
| O1 | 1$b$ | 0.0 | 0.0 | 0.5 | 1.0 | 0.0081 |
| O2 | 1$d$ | 0.5 | 0.5 | 0.5 | 1.0 | 0.0081 |
| O3 | 2$g$ | 0.0 | 0.5 | −0.0023(5) | 1.0 | 0.0081 |
| O4 | 4$h$ | 0.1399(3) | 0.1411(3) | 0.1295(5) | 1.0 | 0.0081 |
| O5 | 4$h$ | 0.1385(3) | 0.3566(3) | 0.6114(4) | 1.0 | 0.0081 |
| O6 | 4$h$ | 0.3619(3) | 0.3608(3) | 0.1239(5) | 1.0 | 0.0081 |
| O7 | 4$h$ | 0.3599(3) | 0.1398(3) | 0.6233(4) | 1.0 | 0.0081 |
| D11 | 4$h$ | 0.0582(3) | 0.0544(4) | 0.3957(2) | 0.5 | 0.0238 |
| D21 | 4$h$ | 0.4450(4) | 0.4430(4) | 0.3950(2) | 0.5 | 0.0238 |
| D31 | 4$h$ | 0.0610(5) | 0.4383(5) | 0.9079(8) | 0.253(2) | 0.0238 |
| D32 | 4$h$ | −0.0587(5) | 0.4438(5) | 0.0989(5) | 0.747(2) | 0.0238 |
| D41 | 4$h$ | 0.2174(5) | 0.2123(5) | 0.1348(2) | 0.5 | 0.0238 |
| D42 | 4$h$ | 0.1709(8) | 0.0547(4) | 0.0509(9) | 0.723(8) | 0.0238 |
| D43 | 4$h$ | 0.1029(9) | 0.1036(9) | 0.2683(7) | 0.5 | 0.0238 |
| D44 | 4$h$ | 0.0556(5) | 0.1717(13) | 0.0461(12) | 0.277(8) | 0.0238 |
| D51 | 4$h$ | 0.2117(6) | 0.2812(6) | 0.6148(23) | 0.343(4) | 0.0238 |
| D52 | 4$h$ | 0.0553(5) | 0.3412(18) | 0.5161(10) | 0.299(4) | 0.0238 |
| D53 | 4$h$ | 0.1045(8) | 0.3930(7) | 0.7530(6) | 0.747(2) | 0.0238 |
| D54 | 4$h$ | 0.1606(10) | 0.4415(4) | 0.5223(8) | 0.614(3) | 0.0238 |
| D61 | 4$h$ | 0.2849(5) | 0.2892(5) | 0.1324(23) | 0.5 | 0.0238 |
| D62 | 4$h$ | 0.3319(8) | 0.4474(4) | 0.0446(9) | 0.785(7) | 0.0238 |
| D63 | 4$h$ | 0.3973(9) | 0.3940(10) | 0.2664(7) | 0.5 | 0.0238 |
| D64 | 4$h$ | 0.4451(6) | 0.3224(11) | 0.0460(14) | 0.215(7) | 0.0238 |
| D71 | 4$h$ | 0.2893(5) | 0.2181(5) | 0.6206(14) | 0.657(4) | 0.0238 |
| D72 | 4$h$ | 0.4501(5) | 0.1609(14) | 0.5474(12) | 0.386(3) | 0.0238 |
| D73 | 4$h$ | 0.3929(13) | 0.1058(13) | 0.7666(7) | 0.253(2) | 0.0238 |
| D74 | 4$h$ | 0.3396(10) | 0.0515(4) | 0.5414(9) | 0.701(4) | 0.0238 |

$a = b = 8.834721\ (33)$ Å.
$c = 5.75542(5)$ Å.
$V = 449.224(4)$ Å$^3$.

sublattice and Bernal-Fowler rules. After analysis through the subgroup tool in BILBAO[40–42], ruling out lower symmetry monoclinic and triclinic space groups as well as centred space groups (by virtue of the observed reflection conditions), a set of 21 higher symmetry space groups is obtained – $P\bar{4}$, $P4_2$, $P\bar{4}2_1m$, $P\bar{4}2m$, $P4_2nm$, $P4_2cm$, $Pbcn$, $Pcca$, $Pnna$, $Pban$, $Pnn2$, $Pna2_1$, $Pba2$, $Pnc2$, $Pca2$, $Pcc2$, $P2_12_12$, $P222_1$, $P222$, $Pmmn$ and $Pmm2$. In order to investigate their compatibility with the O-lattice, the structure was centred in line with their respective transformation matrix[40–42]. This procedure is illustrated in Supplementary Figures 1 and 2 and filters out all space groups, except for five: $P\bar{4}$, $P4_2$, $P2_12_12$, $Pcc2$ and $Pcca$. In order to obtain a feasible set of candidate structures, the H-atom networks of these space groups have to be taken into account. That can be done by observing both Bernal-Fowler rules and the respective space group symmetry. Specifically, each H-site is reflected through all symmetry operations so that H-positions of the same occupancy could be found. The occupancies of even more positions were derived from the Bernal-Fowler rules, as illustrated in Supplementary Figure 3 on the example of space group $Pcc2$, which implies that a structural model in space group $Pcc2$ is necessarily only partially ordered. In total 109 valid structures in five space groups could be determined as possible candidate structure through these considerations, which can still not be tested quantitatively against the data. Moreover, the deviation of these structures from the average ice VI structure is very small and the amount of information required to describe this is represented by just a few quite weak fundamental and super-lattice peaks. Hence, the refinement of structural models is quite delicate, requiring restraints, constraints and damping in order to avoid divergence and to arrive at stable solutions. Even then, the difference in the statistical quality of the fit between models turns out to be small. We have, therefore, based our choice of tested structures on a few specific examples, and once we became aware of the structure fitting reported by Yamane et al.[36], first deposited as a preprint, we have chosen to report the equivalent best-fitting for their four best structural models in space groups, $P\bar{4}$, $Pcc2$ and $P2_12_12$ and $Pca2_1$ (even though the latter can be excluded based on our analysis above). The quality of the fits to the diffraction data are

shown in Fig. 5 (for $P\bar{4}$) and Supplementary Figures 4 (for $Pcc2$), 6 (for $P2_12_12$) and 8 (for $Pca2_1$). These refinements yield accurate atomic coordinates, D-atom occupancies and average isotropic displacement parameters, being reported in Table 1 (for $P\bar{4}$) and Supplementary Tables 1 (for $Pcc2$), 2 (for $P2_12_12$) and 3 (for $Pca2_1$) using the atom labelling as shown in Fig. 6 and Supplementary Figures 5, 7 and 9, respectively. The $P\bar{4}$ model turns out to be the best fit to the neutron powder diffraction data also in our case (see statistical measures for quality of fit summarized in the Supplementary Discussion). For the tetragonal $P\bar{4}$ model a $\chi^2$ of 3.278 was obtained, with $w$Rp = 2.38 %. The tetragonal lattice parameters at 70 K are $a = 8.834721(33)$ Å, $c = 5.75542(5)$ Å, $V = 449.224(4)$ Å³. With respect to an exact $\sqrt{2} \times \sqrt{2} \times 1$ super-cell of ice VI, the volume of ice XIX is smaller by $\Delta V/V = -0.11\%$. This reduction in volume is dominated by a shortening of the $c$-axis ($-0.254\%$) partially offset by a small expansion in the two orthogonal directions perpendicular to $c$.

While the $P2_12_12$ and $Pca2_1$ models clearly fall short of $P\bar{4}$, the $Pcc2$ model results in only a slightly poorer fit to the data. Especially the Bragg peak at 1.87 Å shows the best fit with the $P\bar{4}$ model. For the orthorhombic $Pcc2$ model (see Supplementary Discussion), a $\chi^2$ of 4.018 was obtained, with $w$Rp = 2.63%. The orthorhombic lattice parameters at 70 K are $a = 8.84253(7)$ Å, $b = 8.82654(7)$ Å, $c = 5.75559(5)$ Å, $V = 449.218(5)$ Å³. In common with the tetragonal model, we observe a reduction in volume relative to ice VI of approximately $-0.10$ %, much of which is due to contraction of the $c$-axis. That is, $c$-axis contraction is necessary to reach ice XIX from ice VI, which can be achieved under high-pressure conditions near 2 GPa, but not at low-pressure conditions. Hence the Yamane et al. data obtained in situ at 1.6 GPa and our data obtained ex situ at ~50 mbar can be described using the same structural models and we conclude that their work and ours indicates the same partially ordered superstructure, described as ice XIX. The two best models differ from each other in one important aspect: the two interpenetrating frameworks in the $P\bar{4}$ model are not related by symmetry and we observe that one of the two frameworks becomes substantially more ordered than the other (where site-symmetry constraints limit the occupancy of many H-atoms sites to remain = 0.5). On the other hand, the two frameworks in the $Pcc2$ model are related by symmetry and thus the degree of order achieved is the same in both components of the framework. Since the two leading candidate models crystallise in non-centrosymmetric space groups, a simple powder test (such as second harmonic generation) cannot definitively distinguish between the two choices. On the other hand, $Pcc2$ represents a polar point group, exhibiting pyroelectricity, whereas the $P\bar{4}$ model belongs to a non-polar point group that exhibits piezo-electric properties. Relevant experimental tests to make a distinction on the basis of these properties are hard to do because of the polycrystalline powder nature of our samples. Ideally, we would like to grow single crystals of ice XIX – however, even if we started from a single crystal of ice VI, we do not have a way to avoid the formation of many randomly oriented domains of partly-ordered ice XIX within the ice VI matrix upon cooling. In a very similar manner to the ordering transition of ice VI to yield ice XIX, preferential contraction of the $c$-axis of ice I$_h$ is observed upon transformation to the ferroelectrically-ordered phase ice XI[43]. This leads to local partial ordering of ice I$_h$ on cooling towards the ordering transition and formation of domains of ice XI within ice I$_h$[44].

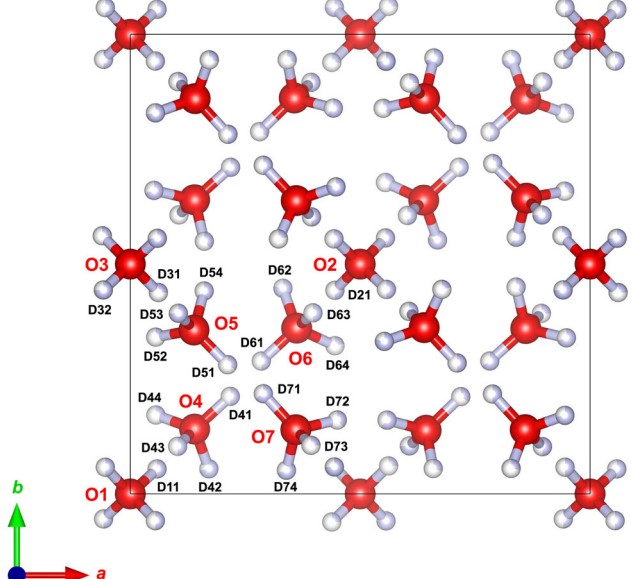

**Fig. 6 Unit cell of ice XIX (refined $P\bar{4}$ model).** View along the $c$-axis, showing the atom labelling scheme used in Table 1. Shading of the hydrogen atoms indicates the occupancy, reported quantitatively in Table 1.

## Discussion

We here report the crystal structure for a partially hydrogen-ordered ice polymorph. Based on common practice, the

availability of the crystal structure allows us to assign a Roman numeral to this ice phase. Following the latest discovery of the superionic phase ice XVIII appearing at ultrahigh-pressure, we here add ice XIX to the ice phase diagram in the versatile intermediate pressure range between 0.5 and 2.0 GPa, in which 13 ice polymorphs are now known. Ice XIX represents an ordered phase sharing the same oxygen atom topology with its parent, disordered ice VI, and its sibling, partially antiferroelectrically ordered ice XV. The tetragonal $P\bar{4}$ structure refines slightly better than the orthorhombic $Pcc2$ model. The difficulty to distinguish between the two is likely to be related to small domains of ice XIX. Indeed, with powder data alone, a conclusive determination of the symmetry resulting from the partial ordering is extremely difficult and efforts to prepare ice XIX as a single crystal are required, as well as systematic computational studies of the thousands of possible orientational configurations. The space group symmetry requires ice VI to be fully hydrogen disordered. Any change in hydrogen order has to be associated with breaking of the space group symmetry[21]. That is, fractional occupancies of D-atoms deviating from it make the case for a new ice structure, as is the case for both structures fitting the data. Ice XIX is slightly better ordered compared to ice XV based on the calorimetry data, where $D_2O$ and $H_2O$ ice XIX release 18% and 14% of the Pauling entropy upon disordering, respectively, compared to 10% in case of $H_2O$ ice XV. Both structures feature cancellation of dipole moments, such that the structure's partial hydrogen ordering is antiferroelectric. That is, our earlier speculation of a possible ferroelectric structure is incorrect[25,26]. Also earlier theoretical predictions about a ferroelectric structure in the $Cc$ space group are incorrect[34,45]. While Nanda and Beran had correctly predicted that the $Cc$ structure is less favourable than the experimental ice XV structure, the supercell necessary to describe ice XIX was not considered[46]. We can certainly rule out any 1×1×1 unit cell, including the one for space group $Cc$, e.g., because our sharp peak at 1.960 Å and the broad peak at about 2.01 Å would be absent in $Cc$. Knight and Singer examined $\sqrt{2}\times\sqrt{2}\times2$ cells and found a nearly energetically degenerate antiferroelectric structure in $P2_12_12_1$[42]. Even though we do not see any evidence for a doubling along c in our ice XIX data, this illustrates how sensitive the ordering is to long-range interactions. And this is where more recent developments in DFT, such as the widespread and easy implementation of dispersion corrections, looks to be essential to solving this very delicate problem. Now that we have established the correct unit cell, it should be possible computationally to provide a robust prediction of the ground-state configuration in future work.

Upon heating ice XIX transforms in two steps first to ice XV above 107 ± 1 K, and then to ice VI at 128 K. This is consistently seen in the heating scans in calorimetry (at 10 K min⁻¹) and neutron powder diffraction scans at HRPD (averaged over the whole run at 2.5 K hour⁻¹). To the best of our knowledge this makes the case for the first order-order transition in the H-subnetwork in a given O-network in ice physics. Ice XIX shows an O-lattice and a partly ordered H-network, making the case for a fully crystalline polymorph, as opposed to a crystal that shows a disordered H-subnetwork. That is, ice XIX, formerly called ice β-XV, cannot be described as a deep glassy state. The mechanism of the order-order transition from ice XIX to ice XV is shown to proceed in an activated fashion, where the activation barrier needs to be determined in future work. In the present work we are able to show that a transient, disordered state is encountered as the transition state – called ice VI‡. The deep glassy state put forward in earlier work by Rosu-Finsen & Salzmann[28,29] could then be the transiently encountered ice VI‡, but not ice XIX. We also rule out the possibility of ice XIX turning into deep glassy ice upon pressure change. The same structural models for ice XIX fit

both data recorded at 1.6 GPa (reported in ref. [36]) and data recorded at 50 mbar (reported in the present work). A comparison of unit cell volumes at 1.6 GPa and 50 mbar translates into an approximate estimate for ice XIX's bulk modulus of 21 GPa at 70 K. This compares with a bulk modulus of 18.5 GPa for its disordered parent ice VI[47] and for ice $I_h$ of 11.2 GPa[7], both at 70 K.

Protiated ice XIX represents the thermodynamically stable phase of water between 1.0 and 1.5 GPa, and below 103 K, as argued by us earlier[25]. That is, ice XIX occupies a substantial fraction of the intermediate pressure regime as a stable phase in water's $p$-$T$ phase diagram. Taking the present results together with the collected work by Salzmann et al. and ourselves on ice XV, it is evident that ice XV forms preferentially at low pressures below 1.0 GPa upon cooling ice VI, whereas for ice XIX high pressures exceeding 1.0 GPa are necessary. At 1 bar ice XV forms, but not ice XIX. At 2.0 GPa ice XIX forms, but not ice XV. At intermediate pressures there is competition between both, where pressure determines the kinetics and the fractions of ice XV and XIX. This is due to the compressed c-axis in ice XIX and the elongated c-axis in ice XV, both compared to the parental ice VI.

## Methods

**Sample preparation.** Sample preparation was done using the same set of equipment as used in our previous work[25]. Ice XIX samples were prepared by (i) cooling 600 µl 0.01 M DCl in $D_2O$:$H_2O$ mixtures (0.04 to 99.99% $H_2O$) to 77 K, (ii) compressing to 1.8 GPa using a piston-cylinder setup with an 8 mm diameter bore in a Zwick BZ100 material testing machine, (iii) heating to 255 K, followed by (iv) slow cooling at rates of 0.1 to 3.0 K min⁻¹ to 77 K. Undoped ice VI was prepared following a similar pathway using a 95:5 $D_2O$:$H_2O$ mixture. After cooling to 77 K ice VI samples were compressed to 1.0 GPa before heating to 255 K, followed by quenching with liquid nitrogen at ≈ 80 K min⁻¹. After releasing the pressure at 77 K, ice XIX and ice VI samples were recovered and stored under liquid nitrogen at atmospheric pressure. The ice XV reference was prepared in situ from ice XIX by heating to 135 K at 50 mbar (converting it to ice VI) and then recooling it with 0.4 K min⁻¹ to 70 K (converting it to ice XV).

**Differential scanning calorimetry.** All quench-recovered ices were analyzed calorimetrically using a Perkin Elmer DSC 8000 at ambient pressure. The samples were encapsulated in aluminium crucibles under liquid nitrogen and transferred to the precooled oven of the calorimeter. Every batch was characterized by heating at 10 K min⁻¹ from 93 to 253 K, which leads to the transformation of both, ice XIX (over the sequence ice VI‡ → XV → VI) and ice VI, to ice $I_h$. The resulting ice $I_h$ was cooled to 93 K and heated in a second scan to 313 K. The second scan was used to correct the background of the first heating run. The enthalpies of fusion of $H_2O$ [48] and $D_2O$[49], 6012 and 6280 J mol⁻¹, were used to determine the mass of ice inside the crucible. The baseline was interpolated between the straight sections before the first and after the second endotherm to evaluate peak areas, onset and offset points. In selected experiments, ice XIX ($D_2O$:$H_2O$ = 95:5) samples were heated from 93 to 120 K at 10 K min⁻¹ and annealed at this temperature for 30 min to allow ice VI‡ to transform to ice XV. Afterwards, the samples were heated to 253 K to record the second endotherm pertaining to the ice XV to ice VI transition.

**Neutron powder diffraction.** Samples of pure $D_2O$ ice VI and doped ice XIX were examined using neutron powder diffraction methods on the High Resolution Powder Diffractometer (HRPD) at the ISIS Neutron and Muon Spallation Source, Rutherford Appleton Laboratory, UK[50]. The various ices were measured in slab-geometry sample holders that are described in detail elsewhere[51], having internal dimensions with width, height and depth = $18 \times 23 \times 5$ mm relative to the incident neutron beam. These were partially assembled and immersed in liquid nitrogen for the loading procedure. The powdered samples were transferred from a nitrogen-chilled steel cryomortar into the sample holder using a nitrogen-chilled spoon. Once filled, the 'back' vanadium foil window of the sample can was attached with screws. Both the 'front' (i.e., beam-facing) and 'back' windows were sealed with 1 mm indium wire to prevent leakage of material from the interior. The front windows and the body of the sample holder were masked with Gd and Cd foil.

The fully-assembled sample holder was then transferred into a top-loading Closed Cycle Refrigerator (CCR) mounted on the HRPD beam line at 70 K. Temperature control was achieved using a cartridge heater and a RhFe thermometer inserted into the aluminium frame of the sample holder, the background temperature of the He exchange gas being maintained ~30 K below the sample temperature where possible. Moving from one temperature to another was done at a rate of 3 K min⁻¹ with a mandatory wait of 10 min at each temperature prior to beginning a measurement in order to be sure that complete thermal equilibrium of the sample was achieved.

Data were collected using a neutron time-of-flight window extending from 30 to 130 ms. In HRPD's highest resolution backscattering detector banks, this TOF range yields data covering $d$-spacings from 0.65 to 2.60 Å, which contains the great majority of Bragg reflections required for a high-precision structure refinement. For example, ice VI has >240 reflections in this $d$-spacing range whereas the next available detector bank at $2\theta = 90 \pm 10°$, which allows us to access $d$-spacings from 1–4 Å, contains only an additional five reflections. Nevertheless, these fewer but more intense peaks are often useful for rapid tracking of phase changes, as shown in Fig. 2.

Raw time-of-flight data were focussed to a common scattering angle for each detector bank, normalised to the incident spectrum and corrected for instrument efficiency by reference to measurements of a V:Nb rod and the empty instrument with the Mantid suite of powder diffraction algorithms[52]. The CCR makes a comparatively small contribution to the background, but of most concern in the case of these data are the weak peaks from the vanadium foil windows used on the CCR tails, as they occur in a region of very weak peaks of interest from the ice XIX sample. Consequently, a long background measurement from an empty masked slab can in the CCR was made at 70 K to ensure accurate subtraction of the parasitic peaks from the structural datasets obtained from ice VI and ice XIX at 70 K.

Structural models in different space groups were fitted to the high-resolution neutron powder diffraction data using the Rietveld method implemented in GSAS/Expgui. Bond length restraints of 0.925 ± 0.015 Å were placed on the O–D bond length, based on the average length found in our ice VI refinements; no bond angle restraints were imposed. Constraints were applied to the isotropic displacement parameters, $U_{iso}$, to achieve an average value for each type of atom (O or D). Constraints were also used to ensure that the D-atom occupancy along each D-bonded O–O vector summed to one. Chemical restraints were imposed to insure that, where not otherwise determined by the occupancy constraints or by symmetry, the total number of D-atoms per O-atom remained fixed equal to two.

## Data availability

Source data are provided with this paper, where all data points in the figures are provided electronically. The ice XIX structure has been deposited with the Cambridge Crystal Structure Database, deposit number 2043863. The high-precision ice VI structure (including the powder diffraction data and model fit) has been deposited with the Cambridge Crystal Structure Database, deposit number 2043864. These data can be obtained free of charge from The Cambridge Crystallographic Data Centre via www.ccdc.cam.ac.uk/data_request/cif.

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

## Acknowledgements

The authors thank the STFC ISIS facility for a beam time allocation, proposal RB2000132 (doi:10.5286/ISIS.E.RB2000132), and the provision of technical resources with which to carry out this work. We gratefully acknowledge support by the Austrian Science Fund (FWF) under grant number I1392.

## Author contributions

T.M.G. and T.L. have designed the study. T.M.G. prepared all samples and characterized them using X-ray diffraction and calorimetry. T.M.G., A.V.T., A.D.F. and T.L. carried out the neutron diffraction work at HRPD. A.D.F. and A.V.T. carried out the space group analysis. A.D.F. did the Rietveld refinement. T.M.G. and T.L. analysed the calorimetry data. T.L. wrote the manuscript with input from T.M.G, A.V.T. and A.D.F.

## Competing interests

The authors declare no competing interests.
