## [Peer Review File · Nature Communications]

REVIEWER COMMENTS

Reviewer #1 (Remarks to the Author):

This work reports evidence for a new form of ice: a partially ordered variant of ice VI, which is crystallographically distinct from ice XV, which is also a partially ordered variant of ice VI. As such, and following the assignment of a crystallographic structure of the new phase, the authors coin it ice XIX. I think that the crystallographic details and thermodynamic information on this new phase of ice will be of interest to the readership of Nature Communications in particular in light of the order-order transition, which is a new phenomenon.

The paper begins with a well written and comprehensive introduction to the topic. This is always appreciated and helpful for the broad community interested in ice. The core of the work then presented is based on the insight that that the introduction of a small amount of H₂O can seed crystallisation of the new phase in bulk D₂O. This latter is critical to enable the high-resolution neutron diffraction data, which cannot be collected from hydrogenous ice.

First, calorimetric data are presented, which convincingly show that even 0.5% seeding of H₂O is enough to result in ordering. I did question the interpretation that the H₂O forms small crystallites that subsequently nucleate the bulk D₂O. Given that one would assume the 0.5% H₂O to be perfectly dissolved in the bulk D₂O, leading to spatially separated molecules of H₂O. How do they then “find each other” to form the nucleating seeds? So, I wondered if some other factor is at play affecting the kinetics. Nevertheless, this seems a detail that doesn't affect the subsequent conclusions of the paper. It is comforting that the differences between the calorimetry in pure H₂O vs H₂O-doped D₂O are well accounted for by understood kinetic effects.

The benefit of having a deuterated form of ice XIX, and access to a high-resolution powder diffractometer are evident from the diffraction patterns shown in their Figure 2. And it seems quite unambiguous that a new phase is formed. Subsequently, the refinements using the reported structure of Yamane et al in ArXiv are mostly convincing. The fact that these data are measured from a large sample at ambient conditions likely mean that these are the best possible data that could have been obtained from a powder sample. The in situ data of Yamane et al are necessarily of lower quality, so it is impressive that they (Yamane) were able to obtain the correct structure from these. The fact that this structure fits the independently measured high-pressure data of Yamane et al, and the recovered data from H₂O-doped D₂O is quite convincing to me that the structural assignment is correct. However, it then raises the question of how Yamane et al were able to crystallise ice XIX without the H₂O dopant.

The Rietveld refinements were certainly difficult to conduct and the authors note that constraints were necessary. I spotted that one of these was to constrain the molecule to have an OD bondlength of 0.925 ± 0.015 Å. This is a very short value, and significantly below that reported by Kuhs et al [J Chem Phys 81 3612 (1984)] of 0.986 Å in ice VI and I think this is worth some comment (short O-D bonds have previously been associated with the presence of multi-site oxygen disorder in ices Ih and VII, is this a factor here?). As the authors state, full confirmation of the structure may require single-crystal techniques and I think this should be a feasible experiment in the future as single-crystal ice VI has been measured in the past by Asbahr and Kuhs in the 1980's.

In summary, I think this is an important work and I would support its publication. I also would highlight the apparently valuable contribution (partially acknowledged by citation in this work) of Yamane et al in a) proposing the space group and b) pointing out the uniqueness of the observation of an ordered-ordered ice phase. On the latter, perhaps the statement of Gasser et al in their penultimate paragraph "This makes the case for the first order-order transition in the H-subnetwork in a given O-network in ice physics", should mention Yamane et al, who had already observed this and stated: "This study first demonstrates the existence of multiple hydrogen-ordered phases for a hydrogen disordered phase" in their own concluding paragraph.

Reviewer #2 (Remarks to the Author):

The authors claim to describe a second hydrogen-ordered phase of ice VI, ice XIX, previously called ice beta-XV by the authors.

The authors provide a well-written, well-founded introduction into the subject of hydrogen-ordered ice phases, which describes well the difficulties to obtain hydrogen-ordered phases in general and more specifically for ice VI, in particular when dealing with the deuterated form, necessary for neutron powder diffraction. The final "trick" is not only doping with DCl, as so necessary for many hydrogen ordered forms, but with a little bit of hydrogenated water on top to provoke nucleation.

The effect of the water addition is impressively well shown by the differential calorimetry scans.

The comparison of the neutron powder diffraction patterns is just amazing, the differences of VI, XV and XIX are obvious, beginning with the size broadening of both hydrogen-ordered phases, and

characteristic peaks being present/absent/shifted in the respective ordered phases. Visual inspection alone reveals a clearly different phase XIX as compared to XV.

Then comes the order-order transition XIX-XV ... which is nicely reported on both, the calorimetry and the neutron diffraction data to cross a disordered ice VI \ddagger phase, not quite identical to ice VI.

What is a bit unclear in the paper, and requires maybe some rewriting is the sample preparation, the preparation of ice XIX. I found it "hidden" at the beginning of the calorimetry section, while I think it merits, it needs a separate short own section, as it is the starting point for both, the calorimetry and the diffraction experiments. I know, there is a detailed section on this in the concluding "Methods" section, but subjectively it was something which disturbed me in the reading flow, not to have read consciously where the starting point of the investigation is.

Concerning the diffraction pattern refinement, I support the argumentation precluding a lower symmetry, an orthorhombic or monoclinic distortion, but rather advocate a more complex description of broader peak shapes. An analysis of the structural situation reveals an enormous number of possible solutions in an handful of possible space groups, too many to be distinguished on the base of only a small number of additional observations as compared to the reference structure of ice VI.

And here comes an original approach: the authors use the outcome of a competing team, preprinted on ArXiv: Both teams let the possible subgroup candidates start kind of a "race". By different criterions. both teams ended up with a choice of five space groups (where the criterion of Gasser et al. is far more solid than the pure figure-of-merit criterion of Yamane et al.). Now, one compares the two race podiums, and finds that only P-4 and Pcc2 are in both... The others, P42, P21212 and Pcca, had less good figures of merit (χ^2) in Yamane et al., although P21212 was really not far away, it should have been checked as well, the same way as Pcc2 had been checked for being able to fit the data.

In the end, just like Yamane et al., the authors come up with two possible solutions of partially ordered ice XIX, P-4 being slightly more likely than Pcc2. It is disappointing (again) not to be able to distinguish even tetragonal and orthorhombic, but the intrinsic peak width does not allow for profiting from the extremely high resolution of HRPD. There is also a topological difference between the two models, which is the interpenetrating two frameworks: in P-4 they are not related by symmetry, and one finds one framework more orders than the other one, while in Pcc2 they are symmetry-related.

It is very unlikely that one will ever achieve to produce a single crystal, as a such should very well finish this ambiguity ... There is a big crystal-physical difference between the two solutions, Pcc2 is polar, P-4 not, so apart from diffraction methods there could be another way to distinguish both (as discussed briefly by Yamane et al.).

Finally, the conclusions are much more than only conclusions, also more than just a summary. It contains a necessary outlook how to possibly solve the remaining ambiguities computationally, and it recalls the extra information this manuscript delivers as compared to the work of Yamane et al.: Ice XIX does not transform upon decompression as speculated by Yamane et al.

My opinion: it should appear in this overdisciplinary journal, not all riddles are solved here, but it is an important finding on an important molecule. Only minor revisions are necessary, if any at all (see above).

Reviewer #3 (Remarks to the Author):

This manuscript reports calorimetric and high-resolution neutron diffraction data which demonstrate the existence of an apparently hydrogen-ordered phase of ice VI which they previously called "beta-ice XV" and now propose to call "ice XIX". Since another ordered form of ice VI has been reported previously (ice XV; Salzmann et al. 2009), the conclusion is that ice VI has (at least) two H-ordered forms. Doping with HCl and a small amount of protonated water appears to be a key element in obtaining this phase from a mainly deuterated liquid.

The data are very convincing and the manuscript is well written and understandable even for a wider public. The authors are known from their previous work on ice and are among the more well-known specialists on this subject. There is no doubt that this paper should be published in some journal, if it is Nature Communications is another question and has to be decided by the editor.

I start with a few more technical issues. In fig. 4 the authors plot the c-axis as a function of temperature to argue why high pressure favors the growth of ice beta-XV over that of ice XV. But such a Clapeyron argument must be based on volume/density, not on a single lattice parameter, unless the parameters are of the relevant phases are strictly identical. Apart from that, the argument requires prior indexing of the unit cell of ice XIX which was not discussed at this stage.

A further issue is how far this new ice phase is in fact “ordered”, or should be called “ordered”. In a perfect world one would hope to measure the ideal change in entropy of $R(\ln 3/2)$ relative to fully disordered ice VI. From the DSC data one should be able to get an estimate on how much heat/entropy is released on heating, but I can’t find a statement on this. If I look at Table I I see that from the 20 H-sites the vast majority is far away from 0 or 1, at least one site is completely disordered (occupancy 0.5). The absence of significant shading of H-atoms in Fig. 6 makes this quite obvious: this phase appears to be essentially H-disordered. It is true that the other form of “ordered” ice VI, ice XV, does not show complete ordering either (PRL Salzmänn 2009). But there the ordering is more convincing, 7 out of the 10 H-sites are within 20% to the perfectly full/empty values, and from data of two cooling speeds it is clear that full ordering can eventually be achieved.

On the same issue of entropy release, a non-specialist wonders why the 0.5% H₂O sample seems to release less heat than the 99.99% sample, assuming that heat release is related to the area below the two curves. The authors argue that this is due to the presence of an intermediate state “VI++” and refer to the Supplementary Material which is a bit frustrating given the importance of this issue.

A minor and less important remark on “state ice V++”: In any phase transition phase 1 goes through a continuous range of states to reach phase 2, and it appears strange to give them a proper name given that these states are unstable and ill-characterized. If I look at the diffraction data of fig. 3, the only evidence for a distinct “state VI++” is a slight shift of the reflection at 1.223 Å, which is probably only detectable after applying the Savitzky-Golay smoothing.

A more fundamental issue concerns the question on the correctness of the structural model they propose for ice XIX, derived from the powder neutron diffraction patterns. The authors admit that there is a huge number of structures within various space groups which all would produce roughly the same diffraction patterns and that the level of precision is not enough to discriminate among these. They then simply propose – in an ad hoc manner - the same space groups which were proposed in a competing manuscript published on ArXiv (Yamane et al.), i.e. their choice is not really based on an independent evaluation but strongly biased by competing work. Note that Yamane et al. are not sure about their choice of space groups either. Taking one of the two space groups they then propose a structural model, but it is obvious that there could be other solutions. To be absolutely clear, this is not the incompetence of the authors but simply the limit of the technique combined with the limit of data quality.

To resume, this manuscript convincingly shows the existence of another ice phase at low temperatures, without doubt the effect of some degree of hydrogen ordering. The paper convincingly determines the unit cell, but the proposed structural model remains highly uncertain. The similarity with a manuscript published on ArXiv (Yamane et al.) is striking and suggests cross-talks prior to the reviewing process.

RESPONSES TO REVIEWER COMMENTS

Reviewer #1 (Remarks to the Author):

This work reports evidence for a new form of ice: a partially ordered variant of ice VI, which is crystallographically distinct from ice XV, which is also a partially ordered variant of ice VI. As such, and following the assignment of a crystallographic structure of the new phase, the authors coin it ice XIX. I think that the crystallographic details and thermodynamic information on this new phase of ice will be of interest to the readership of Nature Communications in particular in light of the order-order transition, which is a new phenomenon. The paper begins with a well written and comprehensive introduction to the topic. This is always appreciated and helpful for the broad community interested in ice. The core of the work then presented is based on the insight that that the introduction of a small amount of H₂O can seed crystallisation of the new phase in bulk D₂O. This latter is critical to enable the high-resolution neutron diffraction data, which cannot be collected from hydrogenous ice.

We wish to thank the reviewer for the work and are grateful for the appreciation of the findings, the praise on the introduction and the recommendation to publish our work in Nature Communications.

First, calorimetric data are presented, which convincingly show that even 0.5% seeding of H₂O is enough to result in ordering. I did question the interpretation that the H₂O forms small crystallites that subsequently nucleate the bulk D₂O. Given that one would assume the 0.5% H₂O to be perfectly dissolved in the bulk D₂O, leading to spatially separated molecules of H₂O. How do they then “find each other” to form the nucleating seeds? So, I wondered if some other factor is at play affecting the kinetics. Nevertheless, this seems a detail that doesn't affect the subsequent conclusions of the paper. It is comforting that the differences between the calorimetry in pure H₂O vs H₂O-doped D₂O are well accounted for by understood kinetic effects.

We are thankful for appreciating our DSC data as convincing. It is correct that we do not envision a nucleus of thousands of H₂O (or HDO) molecules forming first, and then growing D₂O on top of it. This seems very unlikely, as pointed out correctly by the reviewer. Rather the additional H-atoms help speed up the kinetics of the nucleation, presumably by enhancing reorientational dynamics by orders of magnitude. We have already investigated this issue using dielectric spectroscopy in pure D₂O and pure H₂O samples. We are now planning to also study the kinetics of dielectric relaxation in D₂O samples containing small amounts of H₂O, which we will report in a future manuscript. Also future simulation work on this question will be highly appreciated to understand why a tiny bit of H₂O does such a fantastic job in speeding up the ice XIX nucleation.

In order not to convey the wrong impression that a H₂O ice XIX nucleus forms initially we have changed the sentence in the abstract to: “the small H₂O fraction enhances ice XIX nucleation kinetics” and on p.3 to: “Our key result is that nucleation kinetics of deuterated ice β-XV is significantly enhanced by adding small amounts of H₂O.”

The benefit of having a deuterated form of ice XIX, and access to a high-resolution powder diffractometer are evident from the diffraction patterns shown in their Figure 2. And it seems quite unambiguous that a new phase is formed. Subsequently, the refinements using the reported structure of Yamane et al in ArXiv are mostly convincing. The fact that these data are measured from

a large sample at ambient conditions likely mean that these are the best possible data that could have been obtained from a powder sample. The *in situ* data of Yamane et al are necessarily of lower quality, so it is impressive that they (Yamane) were able to obtain the correct structure from these. The fact that this structure fits the independently measured high-pressure data of Yamane et al, and the recovered data from H₂O-doped D₂O is quite convincing to me that the structural assignment is correct.

We agree that our *ex situ* powder data are of higher quality than the *in situ* data presented by Yamane et al. in their simultaneous submission of the ice XIX structure. We also agree that Yamane et al. did a fantastic job in refining somewhat poorer data, but still arrive at the same structural model that also fits our data.

However, it then raises the question of how Yamane et al were able to crystallise ice XIX without the H₂O dopant.

We definitely think that Yamane et al.'s data could be improved significantly by repeating their experiment and adding a little bit of H₂O to the D₂O sample. A large part of their sample is probably still in the disordered ice VI state, with only a small fraction of ice XIX present. By adding H₂O they will be able to increase the ice XIX fraction, and hence quality of diffraction data, significantly. They also used a slightly different strategy to make ice VI; involving much slower cooling (6K/h) and leading to a fine powder of ice VI that has a larger surface area than our cylindrical, compact sample. This latter aspect might work in their favour, leading to a larger fraction of ice XIX forming. Yet, we still think that their fraction of ice XIX in the ice sample is much closer to the pure D₂O sample in our work than in the 95:5 D₂O:H₂O sample. By adding H₂O their ice XIX fraction will definitely increase significantly. So, we think not only the high-pressure cell reduces the quality of their data, but also the fraction of ice XIX within the sample itself.

The Rietveld refinements were certainly difficult to conduct and the authors note that constraints were necessary. I spotted that one of these was to constrain the molecule to have an OD bondlength of 0.925±0015 Å. This is a very short value, and significantly below that reported by Kuhs et al [J Chem Phys 81 3612 (1984)] of 0.986 Å in ice VI and I think this is worth some comment (short O-D bonds have previously been associated with the presence of multi-site oxygen disorder in ices Ih and VII, is this a factor here?).

Regarding O–D bond lengths we tabulate all of the O–D bond lengths from the paper of Kuhs *et al.* (1984) and those from our own ice VI structure refinement below. We emphasize that the error-bars on our values are one order of magnitude smaller than the ones from Kuhs *et al.* (1984), i.e., we have four digits after the comma, as opposed to three. This is mainly due to the fact that their data was collected *in situ* at high pressure conditions (meaning that overlapping parasitic peaks related to the pressure vessel are contained in their data), while ours were collected *ex situ* (after extracting from the high-pressure vessel). They also had no information between 0.65 and 1.55 Å available, which contains 230 reflections, as opposed to the 22 Bragg peaks used by Kuhs et al. for their refinement. Added to this peak overlaps are much reduced, even at short *d*-spacings, because of the very high and nearly *Q*-independent $\Delta d/d$ resolution of HRPD. The modern Rietveld code enables accurate partitioning of intensity between overlapped peaks compared with the method employed by Kuhs *et al.* For this reason, we are confident in the accuracy of the bond lengths for ice VI that we report, and which furthermore form the basis for the bond-length restraints employed in the ice XIX model fits. Yet, considering the large ESDs on the bond lengths of Kuhs *et al.*, there is not a statistically significant difference between the two sets of results. Finally, we note that the ice XIX model refinements of Yamane *et al.* similarly contain some 'short' O–D bond lengths. For example,

their $P\bar{4}$ ice XIX model contains distances as short as 0.740 Å (O2a – D1a) and as long as 1.034 Å (O2c – D4a); there are many O–D lengths around 0.93 Å.

Bond	Multiplicity	Kuhs et al. , 1 GPa/250 K (1984)	Our work (1 bar/70 K)
O1–D2	4	0.986(48)	0.9171(12)
O2–D1	1	0.937(55)	0.9257(16)
O2–D3	1	0.976(51)	0.9358(9)
O2–D4	2	0.939(32)	0.9344(8)
Average*		0.967	0.9248

*Accounting for multiplicity

As the authors state, full confirmation of the structure may require single-crystal techniques and I think this should be a feasible experiment in the future as single-crystal ice VI has been measured in the past by Asbabs and Kuhs in the 1980's.

We agree that better Rietveld refinements will either need a larger fraction of ice XIX or increased domain sizes of ice XIX. Ideally one would like to have a single crystal. While Asbabs/Kuhs were able to make a single crystal of ice VI, this still does not help to make a single crystal of ice XIX. These experiments would require the melting of the ice VI to grow it again from a single seed crystal by slow cooling (a similar method was used by Weir). Starting from a single crystal of ice VI, ice XIX domains will still start to form from several sites, e.g., from several crystal surfaces and within the bulk. Ultimately, this will again result in a crystal containing many domains in different orientations within an ice VI matrix. Just like reviewer #2 we are sceptical whether synthesis of an ice XIX single crystal will be feasible, at least not in the near future. A technique triggering growth of a single domain within the ice VI single crystal would be required – but we are not aware of any technique how to avoid secondary nucleation sites.

In summary, I think this is an important work and I would support its publication. I also would highlight the apparently valuable contribution (partially acknowledged by citation in this work) of Yamane *et al* in a) proposing the space group and b) pointing out the uniqueness of the observation of an ordered-ordered ice phase. On the latter, perhaps the statement of Gasser *et al* in their penultimate paragraph “This makes the case for the first order-order transition in the H-subnetwork in a given O-network in ice physics”, should mention Yamane *et al*, who had already observed this and stated: “This study first demonstrates the existence of multiple hydrogen-ordered phases for a hydrogen disordered phase” in their own concluding paragraph.

Once again we are grateful to the reviewer for emphasizing the importance and uniqueness of our findings. Regarding the novel order-order transition investigated in our work, we would like to point out that Yamane *et al.* do report ice XIX, too. However, Yamane *et al.* do not observe the order-order transition between ice XIX and ice XV, simply because the transformation does not take place *in situ* under high pressure. The ice XIX → ice XV transition is observed exclusively in our *ex situ* study at ambient pressure, though.

Reviewer #2 (Remarks to the Author):

The authors claim to describe a second hydrogen-ordered phase of ice VI, ice XIX, previously called ice beta-XV by the authors. The authors provide a well-written, well-founded introduction into the subject of hydrogen-ordered ice phases, which describes well the difficulties to obtain hydrogen-ordered phases in general and more specifically for ice VI, in particular when dealing with the deuterated form, necessary for neutron powder diffraction. The final "trick" is not only doping with DCl, as so necessary for many hydrogen ordered forms, but with a little bit of hydrogenated water on top to provoke nucleation. The effect of the water addition is impressively well shown by the differential calorimetry scans. The comparison of the neutron powder diffraction patterns is just amazing, the differences of VI, XV and XIX are obvious, beginning with the size broadening of both hydrogen-ordered phases, and characteristic peaks being present/absent/shifted in the respective ordered phases. Visual inspection alone reveals a clearly different phase XIX as compared to XV. Then comes the order-order transition XIX-XV ... which is nicely reported on both, the calorimetry and the neutron diffraction data to cross a disordered ice VI \ddagger phase, not quite identical to ice VI.

We are very pleased about the enthusiastic reaction of reviewer#2 on our findings, using adjectives such as well-founded, well-written, impressively well-shown, or amazing.

What is a bit unclear in the paper, and requires maybe some rewriting is the sample preparation, the preparation of ice XIX. I found it "hidden" at the beginning of the calorimetry section, while I think it merits, it needs a separate short own section, as it is the starting point for both, the calorimetry and the diffraction experiments. I know, there is a detailed section on this in the concluding "Methods" section, but subjectively it was something which disturbed me in the reading flow, not to have read consciously where the starting point of the investigation is.

Thanks for pointing out that the reading flow misses the sample preparation right at the outset of the Results section. We have now included the following sentences as the first ones in the Results section to aid readers in keeping the read-flow: "Both ice XV and ice XIX samples in this work were made through isobaric cooling of ice VI samples, where DCl doping is used to enhance reorientation dynamics. Ice XIX is obtained at 1.8 GPa upon very slow cooling of ice VI from 255 K to 77 K. Ice XV is obtained upon cooling ice VI slowly at ambient pressure from 150 K to 77 K (see Methods section for more details)."

Concerning the diffraction pattern refinement, I support the argumentation precluding a lower symmetry, an orthorhombic or monoclinic distortion, but rather advocate a more complex description of broader peak shapes. An analysis of the structural situation reveals an enormous number of possible solutions in an handfuls of possible space groups, too many to be distinguished on the base of only a small number of additional observations as compared to the reference structure of ice VI. And here comes an original approach: the authors use the outcome of a competing team, preprinted on ArXiv: Both teams let the possible subgroup candidates start kind of a "race". By different criterions. both teams ended up with a choice of five space groups (where the criterion of Gasser et al. is far more solid than the pure figure-of-merit criterion of Yamane et al.). Now, one compares the two race podiums, and finds that only P-4 and Pcc2 are in both... The others, P42, P21212 and Pcca, had less good figures of merit (χ^2) in Yamane et al., although P21212 was really not far away, it should have been checked as well, the same way as Pcc2 had been checked for being able to fit the data.

We are grateful about the support for our crystallographic approach to deduce possible space groups. As suggested by the reviewer structure refinements of models in space-groups $P2_12_12$ and $Pca2_1$ have been included now, and these are reported as Supporting Figures 5 – 8 and Supporting Tables 2 and 3 in the revised manuscript. The Rietveld powder statistics of all four structural models are shown in the summary table of model fit statistics, which is now included in the Supporting Information. This table confirms that $P\bar{4}$ is the best solution and $Pcc2$ remains the second-best candidate, $P2_12_12$ and $Pca2_1$ are less good fits. Hence our conclusions are unchanged and remain in agreement with Yamane *et al.* On top of p.13 the additional refinements are briefly discussed, emphasizing that $P\bar{4}$ is the best solution, by stating “Especially the Bragg peak at 1.87 Å shows the best fit with the $P\bar{4}$ model.”

In the end, just like Yamane *et al.*, the authors come up with two possible solutions of partially ordered ice XIX, P-4 being slightly more likely than $Pcc2$. It is disappointing (again) not to be able to distinguish even tetragonal and orthorhombic, but the intrinsic peak width does not allow for profiting from the extremely high resolution of HRPD. There is also a topological difference between the two models, which is the interpenetrating two frameworks: in P-4 they are not related by symmetry, and one finds one framework more orders than the other one, while in $Pcc2$ they are symmetry-related. It is very unlikely that one will ever achieve to produce a single crystal, as a such should very well finish this ambiguity ... There is a big crystal-physical difference between the two solutions, $Pcc2$ is polar, P-4 not, so apart from diffraction methods there could be another way to distinguish both (as discussed briefly by Yamane *et al.*).

The model fit statistics and examination of especially the fit to the peak at 1.87 Å in Figure 5 and Supporting Figures 4, 6 and 8 suggest that $P\bar{4}$ is the best solution. $Pcc2$ is very close to it, but slightly less good. $P2_12_12$ and $Pca2_1$ are clearly poorer fits than $P\bar{4}$. To do some independent experimental tests based on non-diffraction experiments to distinguish between $P\bar{4}$ and $Pcc2$ is not an easy task. With powder samples it is likely that the best one might hope to achieve would be an independent test of whether the crystals are centrosymmetric or not (for example, SHG). Even so, this does not help distinguish between the most favourable structural models (according to the Rietveld fitting), both of which are non-centrosymmetric. $P\bar{4}$ shows piezoelectricity, whereas $Pcc2$ shows pyroelectricity – these electric properties cannot be easily probed in polycrystalline powders. As the reviewer notes, unambiguous solution of this problem requires a single crystal, which may be very difficult to produce in the ice XIX phase. In the interim, we propose to pursue computational methods, permitting a very broad survey of the possible parameter space and which may also furnish us with useful diagnostic criteria – for example, subtle vibrational features – that can be confirmed experimentally.

Since this discussion is also of interest to the readers of the manuscript, we have included the following sentences on p.13: “Since the two leading candidate models crystallise in non-centrosymmetric space groups, a simple powder test (such as second harmonic generation) cannot definitively distinguish between the two choices. On the other hand, $Pcc2$ represents a polar point group, exhibiting pyroelectricity, whereas the $P\bar{4}$ model belongs to a non-polar point group that exhibits piezoelectric properties. Relevant experimental tests to make a distinction on the basis of these properties are hard to do because of the polycrystalline powder nature of our samples. Ideally, we would like to grow single crystals of ice XIX – however, even if we started from a single crystal of ice VI, we do not have a way to avoid the formation of many randomly oriented domains of partly-ordered ice XIX within the ice VI matrix upon cooling.”

Finally, the conclusions are much more than only conclusions, also more than just a summary. It contains a necessary outlook how to possibly solve the remaining ambiguities computationally, and it recalls the extra information this manuscript delivers as compared to the work of Yamane *et al.*:

Ice XIX does not transform upon decompression as speculated by Yamane et al. My opinion: it should appear in this interdisciplinary journal, not all riddles are solved here, but it is an important finding on an important molecule. Only minor revisions are necessary, if any at all (see above).

We are grateful for the careful reading and the recommendation to publish our manuscript essentially as is in Nature Communications.

Reviewer #3 (Remarks to the Author):

This manuscript reports calorimetric and high-resolution neutron diffraction data which demonstrate the existence of an apparently hydrogen-ordered phase of ice VI which they previously called "beta-ice XV" and now propose to call "ice XIX". Since another ordered form of ice VI has been reported previously (ice XV; Salzmann et al. 2009), the conclusion is that ice VI has (at least) two H-ordered forms. Doping with HCl and a small amount of protonated water appears to be a key element in obtaining this phase from a mainly deuterated liquid. The data are very convincing and the manuscript is well written and understandable even for a wider public. The authors are known from their previous work on ice and are among the more well-known specialists on this subject. There is no doubt that this paper should be published in some journal, if it is Nature Communications is another question and has to be decided by the editor.

We are excited about reviewer #3 being in agreement with the other two reviewers about the convincing data, and well-written and understandable manuscript!

I start with a few more technical issues. In fig. 4 the authors plot the c-axis as a function of temperature to argue why high pressure favors the growth of ice beta-XV over that of ice XV. But such a Clapeyron argument must be based on volume/density, not on a single lattice parameter, unless the parameters of the relevant phases are strictly identical. Apart from that, the argument requires prior indexing of the unit cell of ice XIX which was not discussed at this stage.

The Clapeyron argument is based on entropy data as deduced from our calorimetry experiments and was reported in ref. 23 for the first time in the context of the phase-diagram and the stability region of formerly ice β -XV (now ice XIX). Our discussion of molar volumes in relation to Clapeyron slopes and phase stability are not intended to be connected to the plot of the c-axis temperature dependence. The purpose of the c-axis plot is to show that there are substantial differences between the three phases referred to, ices VI, XV and XIX, and that there are also substantial variations with temperature, which appear to be due to differences in partial order between the phases and perhaps to variations in the degree of order as a function temperature. In order to avoid confusion we have added the sentence "Furthermore, ice XIX (449.2 \AA^3 , see Table 1) also features a smaller unit cell volume than ice XV (450.6 \AA^3)¹⁹." on p.10.

A further issue is how far this new ice phase is in fact "ordered", or should be called "ordered". In a perfect world one would hope to measure the ideal change in entropy of $R(\ln 3/2)$ relative to fully disordered ice VI. From the DSC data one should be able to get an estimate on how much heat/entropy is released on heating, but I can't find a statement on this. If I look at Table I I see that from the 20 H-sites the vast majority is far away from 0 or 1, at least one site is completely disordered (occupancy 0.5). The absence of significant shading of H-atoms in Fig. 6 makes this quite obvious: this phase appears to be essentially H-disordered. It is true that the other form of "ordered" ice VI, ice XV, does not show complete ordering either (PRL Salzmann 2009). But there the

ordering is more convincing, 7 out of the 10 H-sites are within 20% to the perfectly full/empty values, and from data of two cooling speeds it is clear that full ordering can eventually be achieved.

It is certainly true that ice XIX and ice XV are both partially ordered. It is not straightforward to judge on the change of entropy just by inspecting occupancies. In ref. 19 two sets of occupancies are reported, where, e.g., D9 drops from 0.819 to 0.585 and D15 from 0.616 to 0.500 depending on how ice XV is obtained from ice VI. In total 10 D atoms are partially occupied, which is similar to our ice XIX case. Unfortunately, there is no direct way of converting occupancies to entropy changes upon disordering. However, we have made measurements on this issue, which are reported in our ref. 23 and in Fig.1B in the present manuscript. These measurements show that the entropy change is 0.46 and 0.35 $\text{Jmol}^{-1}\text{K}^{-1}$ for H_2O ice XIX and ice XV, respectively, and 0.60 $\text{Jmol}^{-1}\text{K}^{-1}$ for D_2O ice XIX. The former statement is found on p.5 in the manuscript, and has probably been overlooked by reviewer #3. The latter statement has been added on p.5: “From the enthalpy of disordering and the T_{o-d} of 107 K an entropy change of $0.60\pm 0.07 \text{ J/Kmol}$ results. This corresponds to $18\pm 2\%$ of the Pauling entropy.” In order to place this statement also on a more prominent spot, we have added the following sentence to the Conclusions: “Ice XIX is slightly better ordered compared to ice XV based on the calorimetry data, where D_2O and H_2O ice XIX release 18% and 14% of the Pauling entropy upon disordering, respectively, compared to 10% in case of H_2O ice XV.”

On the same issue of entropy release, a non-specialist wonders why the 0.5% H_2O sample seems to release less heat than the 99.99% sample, assuming that heat release is related to the area below the two curves. The authors argue that this is due to the presence of an intermediate state “VI++” and refer to the Supplementary Material which is a bit frustrating given the importance of this issue. A minor and less important remark on “state ice V++”: In any phase transition phase 1 goes through a continuous range of states to reach phase 2, and it appears strange to give them a proper name given that these states are unstable and ill-characterized. If I look at the diffraction data of fig. 3, the only evidence for a distinct “state VI++” is a slight shift of the reflection at 1.223 Å, which is probably only detectable after applying the Savitzky-Golay smoothening.

We absolutely agree that the issue on the release of entropy in comparison between H_2O and D_2O samples is a very important one that deserves more room. While in the present manuscript we focus on the crystal structure of ice XIX, we are currently drafting a manuscript detailing calorimetry data that addresses these very questions, especially the kinetics of the transition in H_2O and D_2O , the intermediate state VI++, as well as the isotope effect on the entropy changes.

As for the name ice VI++, we absolutely agree that there is a continuous range of structures connecting ice XIX and ice XV. However, ice VI++ is not ill-defined, because it represents the single highest-energy structure along this path, which is known as the transition state. According to common practice we describe the transition state by adding a double-dagger to the name of the ice phase, which needs to be ice VI due to the disordered nature of the transition state. This makes the case for ice VI++. This ice VI++ state is detectable also before Savitzky-Golay smoothening, and furthermore also using Raman scattering – which will be outlined by us in a future manuscript on this topic.

A more fundamental issue concerns the question on the correctness of the structural model they propose for ice XIX, derived from the powder neutron diffraction patterns. The authors admit that there is a huge number of structures within various space groups which all would produce roughly

the same diffraction patterns and that the level of precision is not enough to discriminate among these. They then simply propose – in an ad hoc manner - the same space groups which were proposed in a competing manuscript published on ArXiv (Yamane et al.), i.e. their choice is not really based on an independent evaluation but strongly biased by competing work. Note that Yamane et al. are not sure about their choice of space groups either. Taking one of the two space groups they then propose a structural model, but it is obvious that there could be other solutions. To be absolutely clear, this is not the incompetence of the authors but simply the limit of the technique combined with the limit of data quality.

Our structure analysis detailed in Supporting-Figures 1-3 allows exclusion of all space groups, except for five. Reviewer #2 has described this procedure as far more solid than the one chosen by Yamane et al. After this procedure we are still faced with more than 100 structure models in these space groups. As a strategy we have then taken the best structures found by Yamane et al. that are in these five space groups. The $P\bar{4}$ structure fits the data best and in an excellent way, with all other structure falling slightly (Pcc2) or significantly short of the data fit. Of course, someone might come up in the future with a structure model that has even better statistics. However, neither we nor Yamane et al. were able to find such a structure, and so we propose that the $P\bar{4}$ structure model is the correct solution for the ice XIX structure.

To resume, this manuscript convincingly shows the existence of another ice phase at low temperatures, without doubt the effect of some degree of hydrogen ordering. The paper convincingly determines the unit cell, but the proposed structural model remains highly uncertain. The similarity with a manuscript published on ArXiv (Yamane et al.) is striking and suggests cross-talks prior to the reviewing process.

We are thankful for the very positive comments, especially that our main point of the discovery of the new ice phase XIX is convincing. Even more so we appreciate the critical feedback that has helped to improve the manuscript in some places.

DATA AND CODE AVAILABILITY

Our ice XIX ($P\bar{4}$) structure has been deposited with the Cambridge Crystal Structure Database, deposit number 2043863. Our high-resolution ice VI structure (including the powder diffraction data and model fit) has been deposited with the Cambridge Crystal Structure Database, deposit number 2043864. A file containing the raw data underlying all figures is provided in a two Excel files named 'Sourcedata_figures1-3' and 'Sourcedata_refinements'.

REVIEWERS' COMMENTS

Reviewer #1 (Remarks to the Author):

I am mostly happy with the response the authors had to my first set of comments, with a single exception, relating to their use of a constraint of the O-D bondlength to 0.925 Ang.

I had originally referred to Kuhs et al as a reference study for a neutron measurement for ice VI giving O-D to be closer to 0.98 Ang. The authors point out the limitations of this study, due to the available techniques at the time. This is true, but there are many other available studies that show that 0.92 Ang is simply an unrealistically small value for the covalent bond in the water molecule, in any phase of ice, at any pressure.

Here is a small sample of other neutron diffraction studies:

- 1) Floriano et al Nature 329 821 (1987) - ice Ih O-D = 0.983 (5) Ang
- 2) Nelmes et al PRL 81 2719 (1998) - ice VIII O-D = 0.970 Ang
- 3) M-C Bellissent-Funel et al J Chem Phys 87 HDA/LDA ice O-D = 0.97 Ang

There are many other I could draw on, if I had time.

Meanwhile, the bondlength in the free molecule (which must be *shorter* than in the solid state in the absence of H-bonding) has been known to be 0.95 Ang since the 1930's (Barker & Sletor J Chem Phys 3 660 (1935)).

For this reason, I would insist that this must at least be commented on in the manuscript!

Having said all of that, it does not change my belief that this manuscript is very worthy of publication, and I can recommend that it should be accepted.

Reviewer #2 (Remarks to the Author):

The authors reply in a satisfying manner to my previous remarks, therefore I can accept it for publication. Sure, this paper lacks to a certain degree the "originality" of the preceding one of Yamane et al., upon which this work is based to a certain degree. In terms of structure solution, it confirms their structure interpretation (not want to call it "solution"), at provides an additional investigation technique (DSC), it provides neutron diffraction ex situ rather than in situ, the latter lacking sample volume for sufficient, meaningful counting statistics, it provides more detail on the degree of hydrogen-ordering and how to tune it, and they observed first the transition from ice XIX to ice XV.

The work must be published, but, with a little bit less "novelty" than the paper of Yamane et al., I must leave it to the editor to accept it for Nat.Comm. or elsewhere. If technically feasible, it should appear, once Yamane's preprint has appeared in a peer-reviewed journal, and soon after. The authors used even deliberately the outcome of Yamane et al. in their structural investigation, one cannot regard the two papers for that reason completely independently.

Reviewer #3 (Remarks to the Author):

The authors have responded to my queries and have made some amendments, but a few replies are very evasive.

To my query "On the same issue of entropy release.." they simply reply that this issue "is important" and refer to a manuscript which they are currently preparing.

To my query on the correctness of their structural solution they repeat what they already said in the text, and explained that reviewer 2 found their analysis more solid than that of Yamane et al.

To my query on the amount of entropy release they added a sentence which made it clear that this amounts to only 14-18% of the Pauling value and recalled that this is slightly larger than that of ice XV (10%).

This clarification is indeed important since it immediately raises the question on how this phase (and ice XV) can be called "hydrogen-ordered form of ice VI" throughout the text if the entropy change corresponds to only 14-18% of the Pauling value! I think that any reader would agree that the best one could claim is that this phase is "partially hydrogen-ordered", and I strongly advise to correct this throughout the manuscript.

RESPONSES TO REVIEWER COMMENTS

Reviewer #1 (Remarks to the Author):

I am mostly happy with the response the authors had to my first set of comments, with a single exception, relating to their use of a constraint of the O-D bondlength to 0.925 Å.

I had originally referred to Kuhs et al as a reference study for a neutron measurement for ice VI giving O-D to be closer to 0.98 Å. The authors point out the limitations of this study, due to the available techniques at the time. This is true, but there are many other available studies that show that 0.92 Å is simply an unrealistically small value for the covalent bond in the water molecule, in any phase of ice, at any pressure.

Here is a small sample of other neutron diffraction studies:

- 1) Floriano et al Nature 329 821 (1987) - ice Ih O-D = 0.983 (5) Å
- 2) Nelmes et al PRL 81 2719 (1998) - ice VIII O-D = 0.970 Å
- 3) M-C Bellissent-Funel et al J Chem Phys 87 HDA/LDA ice O-D = 0.97 Å

There are many other I could draw on, if I had time.

Meanwhile, the bondlength in the free molecule (which must be *shorter* than in the solid state in the absence of H-bonding) has been known to be 0.95 Å since the 1930's (Barker & Sleator J Chem Phys 3 660 (1935)). For this reason, I would insist that this must at least be commented on in the manuscript!

Having said all of that, it does not change my belief that this manuscript is very worthy of publication, and I can recommend that it should be accepted.

Thank you for the appreciation of our work and apologies for not answering the issue about the O-D bond length to satisfaction in our previous reply. Indeed, 0.917 Å is an unrealistically small value for **ordered** ices (such as ice VIII mentioned by reviewer #1). However, this phenomenon is not unusual for **disordered** ice phases. It not only appears in ice VI, but also in ices III, V (see Lobban et al., now ref.33) and VII (see Kuhs et al., now ref. 30). In the disordered ice VII crystal O-D bond lengths are 0.915 Å *versus* 0.968 Å in the ordered ice VIII crystal. For ice III even O-D distances below 0.9 Å are observed.

The specific example of ice VI was studied computationally by Kuo and Kuhs (now ref. 32) who show that disorder leads to substantial deviations of O-atoms away from their time- and space-averaged positions, the latter being what is sensed in a diffraction experiment. When fitting models using standard crystallographic methods, the result is then an apparent bias in bond lengths and angles - exactly as others and we ourselves report in ice VI.

On p.8 we have added: "Our refined ice VI structure contains comparatively short O-D distances, ranging from 0.917 to 0.936 Å, which is a consequence of orientational disorder leading to static distortions as outlined by Kuo and Kuhs³². The same effect also appears for other disordered ices, such as ices III and V³³ or VII³⁰."

Reviewer #2 (Remarks to the Author):

The authors reply in a satisfying manner to my previous remarks, therefore I can accept it for publication. Sure, this paper lacks to a certain degree the "originality" of the preceding one of Yamane et al., upon which this work is based to a certain degree. In terms of structure solution, it confirms their structure interpretation (not want to call it "solution"), at provides an additional investigation technique (DSC), it provides neutron diffraction ex situ rather than in situ, the latter lacking sample volume for sufficient, meaningful counting statistics, it provides more detail on the degree of hydrogen-ordering and how to tune it, and they observed first the transition from ice XIX to ice XV.

The work must be published, but, with a little bit less "novelty" than the paper of Yamane et al., I must leave it to the editor to accept it for Nat.Comm. or elsewhere. If technically feasible, it should appear, once Yamane's preprint has appeared in a peer-reviewed journal, and soon after. The authors used even deliberately the outcome of Yamane et al. in their structural investigation, one cannot regard the two papers for that reason completely independently.

Thanks, we agree that publication of our manuscript back-to-back with the work by Yamane et al. is a good idea, benefitting the readers.

Reviewer #3 (Remarks to the Author):

The authors have responded to my queries and have made some amendments, but a few replies are very evasive.

To my query "On the same issue of entropy release.." they simply reply that this issue "is important" and refer to a manuscript which they are currently preparing.

To my query on the correctness of their structural solution they repeat what they already said in the text, and explained that reviewer 2 found their analysis more solid than that of Yamane et al.

To my query on the amount of entropy release they added a sentence which made it clear that this amounts to only 14-18% of the Pauling value and recalled that this is slightly larger than that of ice XV (10%).

This clarification is indeed important since it immediately raises the question on how this phase (and ice XV) can be called "hydrogen-ordered form of ice VI" throughout the text if the entropy change corresponds to only 14-18% of the Pauling value! I think that any reader would agree that the best one could claim is that this phase is "partially hydrogen-ordered", and I strongly advise to correct this throughout the manuscript.

Regarding the entropy change the ice XIX and ice XV cases are rather complex and unprecedented. Ice XIX is the only ice polymorph that shows two endotherms and an additional exotherm upon heating. Before delving into the calculation of changes of configurational entropy it, therefore, needs to be clarified how to integrate the peaks. For instance, overlap of the exotherm with the two endotherms reduces the size of the first endotherm – and hence, makes the estimation a lower limit for the real fraction of Pauling entropy. We have used this conservative estimate to obtain the fraction of 14-18% of the Pauling entropy mentioned in the manuscript. The sum of all three peaks on the other hand represents an upper limit. Also the question which onset temperature to use when there are three peaks and the question about thermodynamic equilibrium need to be considered for a careful estimate. Also our finding that different domains of ice XV and ice XIX may form simultaneously adds complexity to the issue. We feel that this level of detail is too specialized for the readers of Nature

Communications and deserves publication in a more specialized journal. In fact, we are currently in the process of finalizing a manuscript on these issues.

Most definitely both ice XV and ice XIX are partially hydrogen-ordered. The new title “Ice XIX” does not contain any reference to hydrogen order to avoid the wrong impression that ice XIX is fully H-ordered. In the abstract and in the main text we make it absolutely clear that we are dealing with partial hydrogen order. This is also evident when inspecting the H-occupancies in Table 1.